# Isolation, Identification, and Biocontrol Potential of Root Fungal Endophytes Associated with Solanaceous Plants against Potato Late Blight (*Phytophthora infestans*)

**DOI:** 10.3390/plants11121605

**Published:** 2022-06-18

**Authors:** Abbas El-Hasan, Grace Ngatia, Tobias I. Link, Ralf T. Voegele

**Affiliations:** Department of Phytopathology, Institute of Phytomedicine, Faculty of Agricultural Sciences, University of Hohenheim, Otto-Sander-Str. 5, D-70599 Stuttgart, Germany; grace.ngatia@uni-hohenheim.de (G.N.); tobias.link@uni-hohenheim.de (T.I.L.); ralf.voegele@uni-hohenheim.de (R.T.V.)

**Keywords:** antifungal endophytes, secondary metabolites, biological control, volatile compounds, anti-oomycete, *Phytophthora infestans*, sporangia germination

## Abstract

Late blight of potato caused by *Phytophthora infestans* is one of the most damaging diseases affecting potato production worldwide. We screened 357 root fungal endophytes isolated from four solanaceous plant species obtained from Kenya regarding their in vitro antagonistic activity against the potato late blight pathogen and evaluated their performance in planta. Preliminary in vitro tests revealed that 46 of these isolates showed potential activity against the pathogen. Based on their ITS-sequences, 37 out of 46 endophytes were identified to species level, three isolates were connected to higher taxa (phylum or genus), while two remained unidentified. Confrontation assays, as well as assays for volatile or diffusible organic compounds, resulted in the selection of three endophytes (KB1S1-4, KA2S1-42, and KB2S2-15) with a pronounced inhibitory activity against *P. infestans*. All three isolates produce volatile organic compounds that inhibit mycelial growth of *P. infestans* by up to 48.9%. The addition of 5% extracts obtained from KB2S2-15 or KA2S1-42 to *P. infestans* sporangia entirely suppressed their germination. A slightly lower inhibition (69%) was achieved using extract from KB1S1-4. Moreover, late blight symptoms and the mycelial growth of *P. infestans* were completely suppressed when leaflets were pre-treated with a 5% extract from these endophytes. This might suggest the implementation of such biocontrol candidates or their fungicidal compounds in late blight control strategies.

## 1. Introduction

Since its earliest epidemic outbreak in the 1840s, late blight of potato incited by *Phytophthora infestans* (Mont.) de Bary has been the most severe biotic constraint threatening potato production worldwide. Under favorable environmental conditions and host susceptibility, the heterothallic fungal-like oomycete can massively produce aerially dispersible sporangia that can either directly germinate or release large numbers of water motile zoospores able to completely destroy not only potato, but also tomato crops within a few days [1,2,3]. The global economic impact associated with late blight on potato in terms of losses incurred due to damage and costs of disease control is conservatively estimated at more than 6 billion USD annually, with losses being heavier in developing rather than developed countries [1,4]. In Sub-Saharan Africa, losses attributed to late blight range from 30–75% [5]. However, when the disease is initiated early in the cropping season losses might increase up to 100% [6].

Over several decades, fighting late blight disease has largely been based on the use of chemicals [7,8,9,10]. Recently, ecologists have succeeded in raising public concern and awareness about the impacts of agrochemicals on the environment [11]. From this perspective, increased public and scientific desire has been elevated to develop alternative environmental-friendly control strategies that might be less harmful to both the consumer and the ecosystem [12,13,14,15]. Introducing beneficial microorganisms as biological control agents (BCAs) might represent a sustainable and reliable solution to replace chemical fungicides in late blight management. Nonpathogenic endophytic microorganisms have been shown to have potential as BCAs in many agricultural systems [16,17]. Endophytic microorganisms may exert their biocontrol activities by producing antimicrobial compounds that suppress plant pathogens or by inducing defense reactions within the plants [18,19]. Moreover, endophytes can also bestow other beneficial properties to their hosts (e.g., nitrogen fixation and/or phytohormone production), which may lead to a reduced use of agrochemicals and maintenance of biodiversity in plant-associated communities [20]. Several attempts regarding the activities of BCAs against the potato late blight pathogen have been reported [21,22]. In 1997, Ng and Webster [23] demonstrated that treating potato foliage with crude extracts of *Xenorhabdus bovienii*, led to a development of late blight symptoms in only 4% of the pathogen-treated plants compared to the untreated control [23]. Daayf et al. [24] established that *Bacillus subtilis* and *Rahnella aquatilis* restricted the growth of *P. infestans* in vitro, reporting inhibitions up to 81%. They also showed that *Serratia plymuthica* could inhibit the growth of the pathogen by >75% on detached potato leaves. In a potted plant experiment, a commercial preparation of *B. subtilis* applied as a foliar spray immediately after *P. infestans* inoculation was effective against late blight, suppressing the disease to below 40% [25]. Loliam et al. [26] demonstrated that *Streptomyces rubrolavendulae* could increase the survival of tomato and chili seedlings from 51.42% to 88.57% and 34.10% to 76.71%, respectively, in *P. infestans*-infested soils. Fungal endophytes isolated from *Espeletia* spp., including *Trichoderma asperellum*, *Aureobasidium pullulans*, *Nigrospora oryzae*, *Chaetomium globosum*, and *Penicillium commune* completely inhibited growth of *P. infestans* in vitro, while *Paecilomyces sinensis* and *Pestalotiopsis disseminate* recorded inhibitions of >60% [27]. Moreover, Gupta [28] also showed that lesion areas caused by *P. infestans* on detached potato leaves were significantly reduced after treatment with spore suspensions of *Trichoderma viride*, *Penicillium viridcatum*, *Myrothecium verrucaria*, or *Trichoderma harzianum* compared to the untreated control. Under controlled conditions, *Trichoderma atroviride* was observed to reduce late blight disease severity by 27% relative to the untreated control in potato plants [29].

Solanaceous plants providing diverse niches for endophytic associations have been shown to harbor a diversity of endophytic fungal species in their leaves, stems, and roots that enhance growth and suppress plant pathogens [14,30,31]. Kim et al. [12] showed that fermentation broths of *F. oxysporum* EF119, isolated from roots of red pepper, controlled late blight by >90% compared to the control in intact tomato seedlings grown under controlled conditions. Andrade-Linares et al. [14] observed that tomato plants colonized by dark septate endophytes isolated from tomato roots recorded enhanced shoot biomass during early stages of vegetative growth. Recently, de Vries et al. [32] have shown that a root endophyte, *Phoma eupatorii*, could suppress mycelial growth of a broad spectrum of *P. infestans* isolates in vitro and also protect tomato plants through the production of anti-oomycete compounds in planta.

Despite several attempts to characterize endophytes successfully controlling late blight disease on potatoes, their effects often did not deliver consistent disease suppression comparable to their chemical counterparts [2,3,4,6]. Due to this lack of consistency of biocontrol activity, research efforts in terms of discovering new bioactive endophytes preferably with multiple modes of action should be accelerated. Hence, the objectives of the present study were (i) to isolate root endophytic fungi associated with four solanaceous plants obtained from diverse regions in Kenya, (ii) to characterize these endophytes according to their phylogeny, (iii) to evaluate their inhibitory effects against *P. infestans* in co-culture, (iv) to explore the modes of action of the most successful fungal endophytes by studying the suppressive effects of their diffusible and volatile metabolites and, (v) to validate the efficacy of these endophytes in combating *P. infestans* in planta.

## 2. Results

### 2.1. Isolation of Endophytic Fungi from Roots of Solanaceous Plants

A total of 357 isolates of fungal endophytes were obtained from roots of potato, tomato, bell pepper, and nightshade from Nyandarua, Kiambu and Kilifi regions in Kenya. The highest number of endophytes were isolated from nightshade (30.5%), followed by potato (25.5%), tomato (23%), and bell pepper (21%) (Figure 1). Among all endophytes, 112 (31.4%) were identified as *Fusarium* spp., 104 (29.1%) readily sporulated on PDA, while 141, accounting for 39.5%, formed no spores on this medium.

### 2.2. Screening of Endophytes for Anti-Oomycete Activity

The results obtained from the primary high throughput screening assay showed that 64 of the total isolates (n = 357) had potential activity against *P. infestans*. However, four of these proved fastidious and ceased to grow in culture while 14 were morphologically identified as *Fusarium* spp. and excluded from subsequent analyses. Among the remaining 46 potentially active isolates, 63% were obtained from Kilifi (Figure 2), with those isolated from bell pepper in this region accounting for 32.6% of the total number of potential antagonists. Other isolates from Kilifi showing potential activity against *P. infestans* were obtained from tomato (17.4%) and nightshade (13%). All antagonistic isolates from Kiambu were from nightshade, representing 15.2% of the selected potential antagonists while those from Nyandarua were isolated from potato and nightshade accounting for 17.4% and 4.3%, respectively. Generally, the number of antagonists isolated per plant species was 15 isolates each for bell pepper and nightshade and eight isolates each for tomato and potato. Interestingly, only nightshade plants were accompanied by potential antagonists from all three regions.

### 2.3. Characterization of the Fungal Endophytes

Based on their ITS sequences, 46 endophytic fungal isolates screened in the dual culture were characterized. The isolates were considered conspecific to species on the NCBI database when their ITS sequences (ITS1-5.8S-ITS4) matched those of the reference with an identity of ≥99% [33]. Based on this criterion, 37 of the 46 sequences were identified to the species level, three isolates were affiliated to higher taxa (phylum or order), while two matched unidentified fungi (Table 1).

The similarities of the remaining four endophytes (NP3S4-63, KA2S1-42, KB2S2-16, and KB2S2-15) did not meet the threshold and showed associations to higher taxa (genus, family, and order). The 37 isolates belonged to 18 species within the 13 genera including: *Albifimbria*, *Aspergillus*, *Myrothecium*, *Cylindobasidium*, *Epicoccum*, *Macrophomina*, *Penicillium, Plectosphaerella*, *Purpureocillium*, *Pyrenochaeta*, *Rhizoctonia*, *Mucor*, and *Colletotrichum*.

The 46 endophytic isolates and their closest BLAST entries form eight distinctive clades, which agrees nicely with their taxonomic identity (Figure 3). Three of the clades represent the two fungal phyla Zygomycota and Basidiomycota, which contain one and three fungal species, respectively. The Zygomycete *Mucor moelleri* was found to belong to the order Mucorales, while the Basidiomycetes could be placed in two orders, Cantharellales (*Rhizoctonia solani*) and Agaricales (*Cylindrobasidium evolvens*), according to Crous et al. [34]. More than 90% of the endophytic isolates were Ascomycetes distributed into three classes, namely Dothideomycetes, Sordariomycetes, and Eurotiomycetes, which fell into four clades (Figure 3). Dothideomycetes and Eurotiomycetes form the bulk of the identified Ascomycetes representing 45.2% and 35.7%, respectively. All Eurotiomycetes identified belong to the order Eurotiales, while the identified Dothiodeomycetes comprise the orders Pleosporales and Botryosphaeriales [34]. Two isolates (KB1S1-4 and KB2S2-15) cluster with the Dothideomycetes, however, they could not be placed in any of the two orders with absolute certainty. Identified Sordariomycetes form the minority of identified Ascomycetes (19%) and are grouped into two orders, Hypocreales and Glomerellales (Figure 3). Unknown isolate KB1S4-9 clustered with Sordariomycetes and was inferred to belong to the Glomerellales as it grouped with species within this order [34]. The closest BLAST match for isolate KB2S2-15 formed the eighth clade (Figure 3).

A deeper look at the distribution of fungal taxa with potential activity against *P. infestans* in relation to the source host plant showed that tomato accommodated two genera, namely *Aspergillus* (87.5%) and *Penicillium* (12.5%), while four genera, with *Pyrenochaeta* being the most abundant (62.5%), were found in potato (Figure 4).

On the other hand, nightshade and bell pepper harbored more diverse fungi giving rise to eight and seven known genera, respectively. Both host plants also harbored unknown species, with the proportion being greater in bell pepper (33.3%) than in nightshade (6.7%).

### 2.4. In Vitro Activity of Endophytic Fungi against Mycelial Growth of P. infestans

In confrontation assays, mycelial growth retardation of *P. infestans* in the presence of one of the 46 fungal endophytic isolates varied significantly (Table 2). *T. harzianum* along with two endophytes (KB1S2-7 and KA1S1-34) suppressed mycelial growth of the pathogen by 84.5%, 78.2%, and 76.5%, respectively. The other endophytes, however, were either only moderate (KB1S4-10 and KT1S1-21), or slight (KT2S2-29 and KB2S2-15) growth inhibitors. Macroscopic observations of the interaction zones showed that some endophytes completely overgrew the pathogen. Members of these endophytes including *T. harzianum*, *Mucor moelleri*, and *Macrophomina phaseolina* recorded the highest inhibition (70.4–84.5%) (Table 2). *Albifimbria terrestris* and *Penicillium simplicissimum* partially overgrew pathogen colonies giving moderate growth inhibition (51.8–56.3%). In other cases, the growth of both endophyte and pathogen stopped once their colonies came into contact. Mycelial growth inhibition in this group was dependent on the growth rate of the endophytes. Fungal endophytes within the genera *Aspergillus*, *Albifimbria*, *Macrophomina*, and *Cylindrobasidium* showed this type of interaction. Interestingly, other endophytes within the genera *Aspergillus*, *Penicillium*, *Purpureocillium*, and *Pyrenochaeta* created inhibition zones with the pathogen. These endophytes showed slight to moderate inhibition percentages (13.3–46.9%) and the inhibition areas created between endophyte and pathogen varied markedly (Table 2). Three unidentified isolates (KB1S1-4, KA2S1-42, and KB2S2-15) formed significantly (α = 0.05) larger inhibition zones measuring 18.8, 18.3, and 14.8 mm, respectively.

### 2.5. Potential of Endophytes Secreting Volatile Organic Compounds Active against P. infestans

Based on the results obtained from dual culture experiments, six endophytic fungal isolates (NA2S2-45, KB2S2-17, KB1S1-4, KA2S1-42, KB2S2-15, and KB2S4-8) were selected and subjected to further characterization regarding their ability to produce volatile organic compounds (VOCs) with anti-oomycete activity. All endophytes were found to produce VOCs with varying activities against mycelial growth of *P. infestans* (Figure 5).

*M. moelleri* (NA2S2-45), *M. phaseolina* (KB2S2-17), and an unidentified Pleosporales species (KA2S1-42) retarded the growth of the pathogen to a similar extent, recording inhibition percentages of 56.7%, 50.8%, and 48.9%, respectively. Lower suppression levels were observed in the case of *Alibifimbria terrestris* KB1S4-8 (18.4%) and two unidentified fungi, KB1S1-4 (26.8%) and KB2S2-15 (16.7%).

### 2.6. Effect of Crude Extracts from Selected Endophytes on Sporangial Germination of P. infestans

To assess the potential of crude extracts (CEs) obtained from selected fungal isolates against *P. infestans*, sporangia were allowed to germinate in the presence of 5% of individual CEs and germination was evaluated after 16 h. Sporangia germination in 5% acetone (solvent control) recorded ≥90% and differed only insignificantly from that of the water control. Sporangia reacted differentially to the individual CEs applied (Table 3).

While the crude extract (5%) from NA2S2-45 significantly retarded sporangia germination (30%), no suppressive effect on germ tube growth was detected. A similar dosage of the crude extract from KB1S1-4 suppressed sporangial germination by approximately 70% and diminished germ tube elongation by >80% compared to the solvent control. The amendment of *P. infestans* sporangia with CEs obtained from two endophytic fungi (KB2S2-15 and KA2S1-42) entirely suppressed their germination (Table 3). As a result, CEs from KB2S2-15, KA2S1-42, and KB1S1-4 showed suppressive effects against sporangia germination and germ tube development and hence were selected for further in vivo investigations.

### 2.7. In Vivo Activity of the Crude Extracts against P. infestans

To validate the in vitro results, the effect of CEs on leaf blight development on detached potato leaves was investigated. CEs were incorporated into sporangial suspension to give a dosage of (5%, w:v). This mixture was inoculated on the abaxial surface of detached leaflets on either side of the midrib. The results obtained from two independent experiments revealed that detached leaflets treated with either acetone or CEs in the absence of the pathogen did not show any phytotoxic damage (Figure 6b,d–f).

Under a high relative humidity, necrotic lesions covered with white fluffy hyphae and sporangia of *P. infestans* developed on the inoculated leaflets treated only with water (control) within 7 d (Figure 6A). Similarly, no decrease in lesion development or mycelial colonization was detected on inoculated leaflets treated with 5% acetone compared with the water control treatment (Figure 6B). In contrast, late blight symptoms and mycelial growth were completely suppressed when the systemic broad-spectrum fungicide Infinito^®^ was applied (Figure 6C). Interestingly, the application of 5% CEs of the endophytes KB1S1-4, KA2S1-42, and KB2S2-15 yielded neither visible lesions nor hyphal growth of *P. infestans* on the inoculated leaflets (Figure 6D–F).

## 3. Discussion

The screening and identification of microorganisms with antagonistic properties against soilborne pathogens are indispensable first steps in the search for potential biocontrol agents [27,33]. In the current study, root endophytic fungi were isolated from four solanaceous plant species obtained from three regions in Kenya with diverse climatic conditions and soil properties (Table 1). This might increase the heterogeneity and number of fungal species obtained, and hence the possibility of identifying endophytes with unusual adaptation strategies and potential bioactivity against *P. infestans* [27]. The sampling regions were a few to several hundred kilometers apart (1000 km maximum distance), differed in elevation (2532 m maximum difference), climate, and physio-chemical soil properties (Appendix A).

In addition to soil and climatic factors, host plant species also influence the composition of root microbiomes, with roots being shown to harbor more endophytic biodiversity than the rest of the plant organs [35,36]. These factors may have contributed to the realization of 357 endophytic fungal isolates from Kilifi, Nyandarua, and Kiambu from the roots of four solanaceous plant species sampled, namely *S*. *tuberosum*, *L*. *esculentum*, *S*. *nigrum*, and *C*. *annuum*. However, only a limited proportion of approximately 13% of the isolates showed potential activity against *P. infestans*. Interestingly, 63% of the endophytes were obtained from Kilifi (Figure 2), a non-potato growing region, implying that spatial separation could be a predominant factor limiting pathogen–antagonist interactions. Similar findings of the occurrence of unique endophytic fungal species from Kilifi were reported by Bogner et al. [37]. These authors attributed their findings to the hot and humid climate experienced in the coastal region.

The type of host plant may also have been an influential factor in the availability of antagonists. Bell pepper and nightshade each gave rise to the largest number of antagonistic endophytes (Figure 2), which was consistent with the finding of Kim et al. [12], that red pepper roots harbored endophytic fungi with potent activity against *P. infestans*. The endophytes from nightshade and bell pepper were significantly more diverse than those from other host plants (Figure 4). In these two plant species, unidentified endophytes were also captured, all isolated from Kilifi. The diversity of antagonists was lower in potato and least in tomato (Figure 4).

Although challenges in the use of sequence information on the NCBI database for molecular identification of fungi have been cited, it remains a useful tool in the identification of species, especially for non-sporulating endophytes [27,38]. In the current study, there was considerable agreement between the phylogenetic relationships of the endophytes and their corresponding closest BLAST matches, as demonstrated by the high bootstrap support at the terminal nodes (Table 1 and Figure 3). The exceptions were isolates KB2S2-15 and KA2S1-42, for which BLAST matches with only little similarity were found that did not cluster in the phylogeny. These endophytes were non-sporulating on different nutrient media, hampering morphological characterization. Similarly, isolates KB1S1-4, KB1S4-9, KB2S2-16, and KB2S4-18 were also non-sporulating and their identity remained concealed, as their closest BLAST matches were unknown fungal species. The isolates KB1S1-3, NP2S1-60, and NP3S4-63 could not be identified to the species level, but molecular and morphological traits established their genera as *Aspergillus*, *Pyrenochaeta*, and *Periconia*, respectively. On a broader perspective, most of the antagonistic endophytes characterized in this study were Ascomycetes, while the occurrence of Basidiomycetes and Zygomycetes was limited to one or a few isolates. The Ascomycota is the largest and most diverse fungal phylum, and its members are usually the dominant endophytes in plants in relation to other fungal groups [39,40,41,42].

By observing the interactions of 46 potential fungal endophytes in dual culture with *P. infestans*, some antagonists rapidly grew over the pathogen colony resulting in the highest inhibition percentages. Apart from *M. moelleri*, this group comprises mainly of members of possible potato pathogens. While the pathogenicity of these endophytes has not been yet verified, *M. moelleri* has been recently indicated as a potential antagonist against different tomato pathogens including *Athelia rolfsii* and *Colletotrichum gloeosporiodes* [43]. *T. harzianum* T16 also showed a similar interaction with *P. infestans*. Previous reports of the activity of *Trichoderma* spp. against *P. infestans* and other pathogens have demonstrated their capacity for complete growth over pathogen colonies in dual culture, mycoparasitism, and antibiosis through the production of active compounds such as 6-pentyl-α-pyrone, viridiofungin A, harzianolide, and harzianic acid [44,45,46,47]. In exclusion of putative potato pathogens, *Aspergillus* spp., *Penicillium* spp., *Albifimbria terrestris*, and *Cylindrobasidium evolvens* were among the antagonistic fungal endophytes identified. Members of *Aspergillus* and *Penicillium* have been shown to have endophytic lifestyles in plants as well as exhibiting bioactivity against plant pathogens (e.g., *Botrytis cinerea*, *Trichothecium roseum, Sclerotinia sclerotiorum, Fusarium oxysporum*, and *Rhizoctonia solani*) with *A. niger*, *A. terreus*, and *P. citrinum* being some of the well-studied antagonists [33,39,42,48,49]. In addition, a protein derived from *Aspergillus giganteus* was shown to have activity against *P. infestans* in vitro [50]. The Basidiomycete *C. evolvens*, as a saprophyte capable of infecting wounded stem and root tissue, is associated with decay in woody plants [51,52]. In this study, *C. evolvens* was isolated as an endophyte from the roots of *C. annuum* and showed a moderate antagonistic activity against *P. infestans* in dual culture (Table 2). Likewise, *A*. *terrestris* was isolated from the roots of *C. annuum* and is morphologically similar to *A*. *verrucaria*, whose basionym was *Myrothecium verrucaria*, a known plant pathogen that has been formulated into bioherbicides or nematicides [53,54,55]. Interestingly, some endophytes exerted their antagonism by antibiosis as indicated by the inhibition zones. In a comparable study, CEs of *Aureobasidium pullulans* isolated from *Espeletia* spp., a native Andean plant, showed in vitro activity against *P. infestans* [27].

In addition, endophytic fungi are also capable of producing VOCs. VOCs from *M. moelleri*, *M. phaseolina*, and an unknown endophyte KA2S1-42 showed a higher mycelial growth inhibition of *P. infestans* than those released by KB1S1-4, *A. terrestris*, and KB2S2-15 (Figure 5). Previous reports have shown that *Mucor* spp. are able to produce ethanol while *M. phaseolina* has been associated with the production of several fatty acid methyl esters with an inhibitory activity against *Sclerotinia sclerotiorum* [42,56]. On the other hand, *A. verrucaria* excreted bioactive compounds, including antibacterial cyclopeptides that showed herbicidal activities as well [57,58]. Moreover, the *A. verrucaria* isolate SYE-1 exhibited a wide antifungal activity against *B. cinerea*, *Lasiodiplodia theobromae*, and *Elsinoë ampelina* on grapevine [59]. *Myrothecium inundatum*, a close relative of *A. terrestris*, has been implicated in the production of a variety of hydrocarbon VOCs, or their derivatives, potent against *Pythium ultimum* and *S. sclerotiorum* [60].

VOCs have been indicated to have a greater coverage of soil and organic substrata than diffusible organic compounds (DOCs), a characteristic attributed to their gaseous state [61]. In the current study, three unknown antagonists KA2S1-42, KA1S1-4, and KB2S2-15 showed the capability of producing VOCs as well as DOCs that affect *P. infestans*. In addition, *M. moelleri* showed a high inhibition of *P. infestans* mycelium both in dual culture and through the production of VOCs. These findings led to the selection of the antagonistic endophytes KA1S1-4, KB2S2-15, KA2S1-42, and *M. moelleri* for further testing while putative pathogens were left out of subsequent studies.

Consistent with dual culture findings, CEs from the antagonistic endophytes KB1S1-4, KB2S2-15, and KA2S1-42, showed a suppressive activity against sporangia germination and the germ tube growth of *P. infestans*. While CEs from KB1S1-4 at a concentration of 5% exhibited considerable suppressive effects against sporangial germination and germ tube elongation of the pathogen, CEs from KB2S2-15 and KA2S1-42 entirely blocked sporangia germination (Table 3). In a similar set up, Linkies et al. [62] found that upon exposure to ethyl-acetate extracts from *Chaetomium cochliodes* and *C. elatum* at a dosage of 30%, sporangia germination of *P. infestans* was considerably inhibited. However, this dosage is much higher (six-fold) than that used in our study.

On the other hand, although CEs from *M. moelleri* (NA2S2-45) showed a relatively weak activity against sporangia germination, it did not inhibit germ tube development, suggesting that the anti-*Phytophthora* metabolites produced by this fungus are rather confined to the gaseous phase, which could not be extracted from the culture filtrates in concentrations sufficient to affect the germination of *P. infestans* sporangia. Several reports agree with the ability of endophytic, rhizospheric, or phyllospheric fungi in producing diffusible compounds with anti-oomycete activity [12,27,45,63,64]. Tellenbach et al. [65] reported that a strain of *Phialocephala europaea* was able to produce metabolites containing sclerin and sclerotinin with activity against *Phytophthora citricola*. Our earlier studies showed that strain T23 of *T. harzianum* (recently re-identified as *T.*
*asperellum*) secreted viridiofungin A with a strong inhibitory effect against sporangia germination of *P. infestans* [45]. In the current study, although the biologically active metabolites produced by KB1S1-4, KB2S2-15, and KA2S1-42 have not been identified, it was evident that these endophytes produce potent compounds with a strong activity against *P. infestans*.

In the detached leaflet assay, the application of CEs from the endophytes KB1S1-4, KB2S2-15, and KA2S1-42 entirely protected inoculated leaflets and suppressed the hyphal growth of *P. infestans* (Figure 6). Likewise, Bae et al. [66] found that CEs from *Trichoderma atroviride* showed inhibitory activities against *Phytophthora sojae, P. capsici*, and *P. melonis* and induced defense reactions in the detached leaves of pepper and tomato plants. Furthermore, Kim et al. [12] reported a substantial in vivo activity of *Fusarium oxysporum* strain EF119 against tomato late blight. Similar results were also reported by Chandrakala et al. [64], where culture filtrates from *T. virens* and *T. viride* proved to be effective against sporangia germination of *P. infestans* and impeded the establishment of late blight on potato.

Overall, we report preliminary evidence for targeting the late blight pathogen by particular fungal endophytes both in vitro and in vivo. Based on their promising in vitro performance against the late blight pathogen, three endophytes (KA1S1-4, KB2S2-15, and KA2S1-42) were selected from a total of 357 isolates. These endophytic isolates proved to be very active in protecting potato leaflets from *P. infestans*. This biological activity is associated with DOCs and VOCs with anti-oomycete properties. However, the purification and elucidation of the chemical structures of these bioactive molecules have not been accomplished in this study. Therefore, it is still a rich field for future investigations, particularly in terms of clarifying modes of action of the metabolites involved to determine whether they target single or multiple pathways in *P. infestans*. This will shed light on the possibility of combining them to improve their efficacy and hence both the reliability and durability of the biocontrol.

## 4. Materials and Methods

### 4.1. Sampling Regions and Collection of Plant Materials

Root samples from four solanaceous plant species including potato (*Solanum tuberosum* L.), tomato *(Lycopersicon esculentum* L. Mill.), bell pepper (*Capsicum annuum* L.), and African nightshade (*Solanum nigrum* L.), were collected from three sampling regions in Kenya (Nyandarua, Kilifi, and Kiambu) with variable soil types. From each region, root systems of twelve apparently healthy plants were obtained along with representative soil samples. Immediately after collection, intact roots were washed thoroughly under running tap water to remove soil and adhering debris, air dried, and stored at 4 °C until further processing. In order to determine their physio-chemical properties, soil samples were analyzed at the Soil Science laboratories of Jomo Kenyatta University of Agriculture and Technology (Juja, Kenya) (Appendix A). pH and electrical conductivity (EC) were determined in a soil: water mixture (2:5, w:v), while total soil organic carbon was measured using the Walkley–Black rapid titration method [67].

### 4.2. Isolation of Fungal Root Endophytes and P. infestans

Root segments of 3–4 cm from primary and secondary roots of each plant were surface sterilized by immersion in 70% alcohol for 1 min and immediately transferred into 2.5% sodium hypochlorite (Carl Roth, Karlsruhe, Germany) for 5 min. Root segments were rinsed three times with sterile distilled water for 5 min and blotted between sterile paper towels to remove excess moisture. To ascertain the success of surface sterilization, sterilized root segments were briefly imprinted on PDA (Appendix A). Subsequently, the ends of the root segments were trimmed off and further divided into three fragments of approx. 1 cm length, which were placed on PDA amended with 0.05 g/L chloramphenicol (Merck KGaA, Darmstadt, Germany). Finally, plates were sealed with Parafilm and incubated at 22 ± 1 °C in the dark. Only endophytic fungi emerging from successfully surface sterilized roots (Appendix A) were subcultured onto fresh PDA. Isolates were purified through either single-spore isolation or hyphal tip transfer for sporulating and non-sporulating colonies, respectively. Pathogens (e.g., *Fusarium* spp.) were excluded from further tests.

*P. infestans* was isolated from single lesions of potato leaves obtained from a late blight susceptible variety (Duke of York) growing under field conditions in Hohenheim, Stuttgart, Germany. To induce fresh sporulation, blighted leaves were placed in a humid chamber at 20 °C for 48 h. Colonized leaf segments were placed between two surface sterilized tuber slices (5 mm thickness) and incubated for 7 d at 20 °C in the dark (Appendix A). Mycelia emerging from the upper side of a slice were transferred to unclarified V8-based agar medium containing 200 mL V8 juice, 2 g CaCO_3,_ 0.05 g β-sitosterol, 0.05 g ampicillin, 0.05 g vancomycin, 0.01 g pentachloronitrobenzene (PCNB), and 15 g agar in 800 mL deionized water. A pure culture of the pathogen generated from the tip of a single hypha was maintained on corn meal agar (CMA, 17 g/L; Sigma-Aldrich, Munich, Germany) and reactivated after every 3–4 transfers on CMA by inoculation on sterile potato leaflets.

### 4.3. Primary Screening of Endophytes for Anti-Oomycete Activity against P. infestans

A high throughput confrontation assay was established to screen 357 potential antagonistic endophytes against *P. infestans*. Owing to the slow growth of *P. infestans*, 5 mm Ø agar plugs from the border of an actively growing colony were inoculated on the center of 20% V8 agar plates and incubated at 20 °C for 72 h in darkness. Subsequently, agar plugs (5 mm Ø) from four different endophytic fungi were placed equidistant, 5 mm from the edge of the Petri plates pre-inoculated with the pathogen (Appendix A). Control treatments were set up in a similar way in the absence of endophytes. Isolates observed to retard the growth of *P. infestans* compared to the control were selected for further testing and purified to generate axenic cultures.

### 4.4. DNA Extraction, PCR Conditions, and Sequencing

Genomic DNA was extracted from mycelia of root endophytic fungi selected from the screening experiment using the method described by Liu et al. [68]. PCR was conducted in a 40 μL reaction mixture using ITS1 (5′-TCCGTAGGTGAACCTGCGG-3′) and ITS4 (5′-TCCTCCGCTTATTGATATGC-3′) primers [69] to amplify the internal transcribed spacer regions (ITS1-5.8S-ITS2). A single PCR reaction was comprised of 8 µL of Phusion^®^ HF buffer (5×), 0.8 µL of dNTPs (10 mM), 1 µL of each of the forward and reverse primers (10 µM), 0.4 µL of Phusion^®^ polymerase (2 U/µL), and 27.8 µL of ultra-pure water. The PCR program started with an initial denaturation step at 98 °C for 30 s, followed by 35 cycles at 98 °C for 10 s, 54 °C for 20 s, and 72 °C for 35 s and a final extension step at 72 °C for 10 min. Success of amplification was ascertained on 1% agarose gels visualized with 0.05% ethidium bromide with the aid of a gel documentation system (Quantum 1100 PEQLAB, VWR, Darmstadt, Germany). Amplified PCR products were purified using Innu PREP PCR-pure kit (Analytik Jena AG, Jena, Germany) and their concentration adjusted to meet requirements for Sanger sequencing by Source Bioscience Company (Berlin, Germany). Sequences from single reads with the forward primer (ITS1) were trimmed and edited with GENtle v 2.0 and compared to those deposited at the NCBI database. Endophytic fungi whose sequences showed a similarity of >99% to database entries were considered identical to the reference fungi [33], and their taxonomic classification carried out using the MYCOBANK Database [34]. Evolutionary history was deduced using the Neighbour-Joining method as described by Saitou and Nei [70] with a bootstrap test of 1000 replicates to determine the percentage when replicate trees of related taxa clustered together [71]. The maximum composite likelihood method [72] was used to calculate the evolutionary distances, used to infer the phylogenetic tree. Finally, sequence data were submitted to GenBank.

### 4.5. Establishment the Anti-Oomycete Activity of the Endophytes

To validate the inhibitory effect of screened fungal endophytes against *P. infestans*, a dual culture assay was performed. The assay was similar to that described under 4.3 except that the pathogen and each unique endophytic isolate were inoculated at 7 cm distance. *P. infestans* was also inoculated on 20% V8 agar plates 72 h prior the antagonistic isolates. Three replicates were prepared for each isolate and plates inoculated only with the pathogen served as control. *Trichoderma harzianum* (T16) shown to have antagonistic properties against several plant pathogens [45,46] was used as a positive control, while pathogen colonies grown in the absence of the endophytes served as negative control. The experiment was laid out in a randomized complete block design. The initial (72 h post inoculation; hpi) and the final (12 d post inoculation; dpi) colony diameter of the pathogen were measured. The percentage of mycelial growth inhibition was calculated using the formula ascribed by Edgington et al. [73]. In addition, the inhibition zones created between the pathogen and the tested endophytic fungal colonies were recorded.

### 4.6. Impact of Volatile Organic Compounds Extracted from the Endophytes on Mycelial Growth of P. infestans

The ability of selected endophytic fungi to produce volatile organic compounds (VOCs) was assessed following the modified procedure of El-Hasan et al. [74]. Briefly, agar plugs (5 mm Ø) bearing *P. infestans* mycelium were placed on the center of V8 agar plates and incubated for 72 h. Subsequently, the lid was replaced with an upside-down agar plate inoculated with an agar plug colonized by an endophyte. Both Petri dishes were separated with a sterile cellophane sheet and held together with Parafilm. Positive and negative controls consisted of *T. harzianum* and sterile PDA plugs, respectively. Mycelial growth inhibition (in %) was calculated as described above.

### 4.7. Suppressive Activity of Crude Extracts of Root Endophytic Fungi against Sporangial Germination of P. infestans

Putative endophytic fungi that formed inhibition zones in dual culture and/or produced active VOCs were tested for their ability to produce diffusible organic compounds with anti-oomycete activity. To this end, 1 L of 20% V8 broth was inoculated with ten agar plugs (5 mm Ø) of an actively growing endophyte culture. The cultures were incubated on an orbital shaker at 125 rpm and 20 °C in the dark for 12 d. Control treatments, where the endophytes were absent, were set up. Crude extracts (CEs) were prepared from culture filtrates according to El-Hasan et al. [45]. The resulting CE residues were re-dissolved in 2 mL acetone.

Sporangia of *P. infestans* were produced on detached potato leaflets. For this purpose, an agar plug (3 × 3 mm) was placed on the abaxial side of a detached leaflet surface sterilized with 2.5% sodium hypochlorite. A drop of sterile deionized water was introduced on the interface and the inoculated leaflets were placed in a humid chamber made by inverting the Petri dish containing water agar over lids containing sterile moist filter paper. Cultures were incubated at 20/18 °C (light/dark) for 7 d in a growth chamber with a 16 h light cycle. Sporangia were harvested by briefly vortexing infected leaflets in a sterile centrifuge tube containing V8 liquid medium. Mycelia and leaf fragments were trapped using double layers of sterile muslin cloth and the resulting sporangial suspension (5 × 10^4^ sporangia/mL) was used in subsequent experiments.

To determine the effect of CEs on sporangia germination, 25 µL of each extract were combined with an equal volume of water in a sterile 1.5 mL microcentrifuge tube. The tubes were aseptically left open for three hours to allow solvent evaporation. Subsequently, 475 µL of the sporangia suspension were added to the CE solution. Cultures were incubated at 20 °C in the dark to allow sporangia germination, the experiment was terminated after 16 h by the addition of 100 µL lactophenol blue. Acetone was used as solvent control. The percentage of germinated sporangia and the length of germ tubes were determined using a light microscope (Axioskop 2, Carl Zeiss Microscopy GmbH, Göttingen, Germany). Images were taken with an AxioCam MRm digital camera (Carl Zeiss) and measurements determined with the corresponding AxioVision software (SE64 Release 4.8.3 SP1; Carl Zeiss).

### 4.8. Activity of the Crude Extracts against P. infestans on Detached Potato Leaflets

For the in vivo bioassay, leaflets were harvested from the fourth to sixth fully expanded leaves obtained from potato plants (var. Duke of York) grown under greenhouse conditions. CEs that showed activity against germination of *P. infestans* sporangia were tested for their activity in vivo against the pathogen on detached leaflets. Acetone was evaporated from the crude extracts as described above, and each extract reconstituted to give a concentration of 5% in a sporangial suspension. Two droplets of 50 µL each were inoculated on the abaxial surface of detached leaflets on either side of the midrib. Control treatments were set up in a similar way and comprised water, 5% acetone, and a conventional fungicide (Infinito^®^; Bayer CropScience, Langenfeld, Germany). Leaflets were placed individually in a humid chamber and incubated for 7 d at 20/18 °C (light/dark) with a 16 h photoperiod. All treatments were set up in four replicates. The assay was conducted twice.

### 4.9. Data Analysis

If not mentioned elsewhere, experiments were repeated at least twice in triplicate in a completely randomized design. All statistical analyses were performed using SAS software (SAS Institute Inc.). The MIXED and GLIMMIX procedures were applied to determine whether there were treatment effects in the experiments conducted within this study. Data were assessed and transformed when necessary to ensure they met model assumptions. Pairwise comparisons among treatments were carried out using Tukey test (α = 0.05).

## Figures and Tables

**Figure 1 plants-11-01605-f001:**
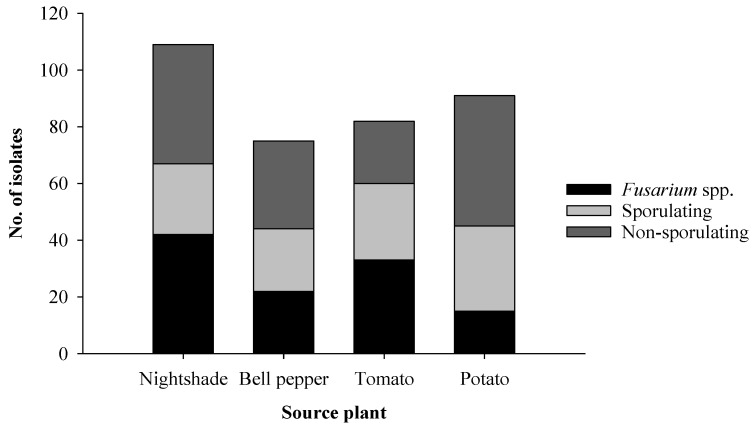
Distribution of the total fungal root endophytes found in four solanaceous host plants and classified into three categories: fusarial, sporulating, and non-sporulating endophytes.

**Figure 2 plants-11-01605-f002:**
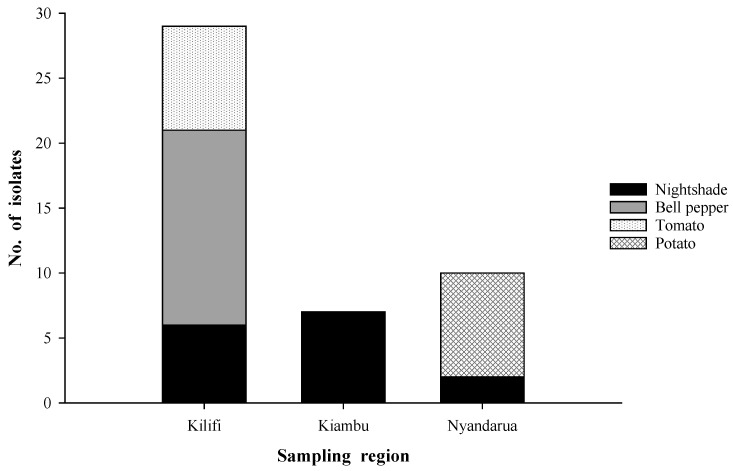
Distribution of root fungal endophytes with potential activity against *P. infestans* in relation to host plant and sampling region.

**Figure 3 plants-11-01605-f003:**
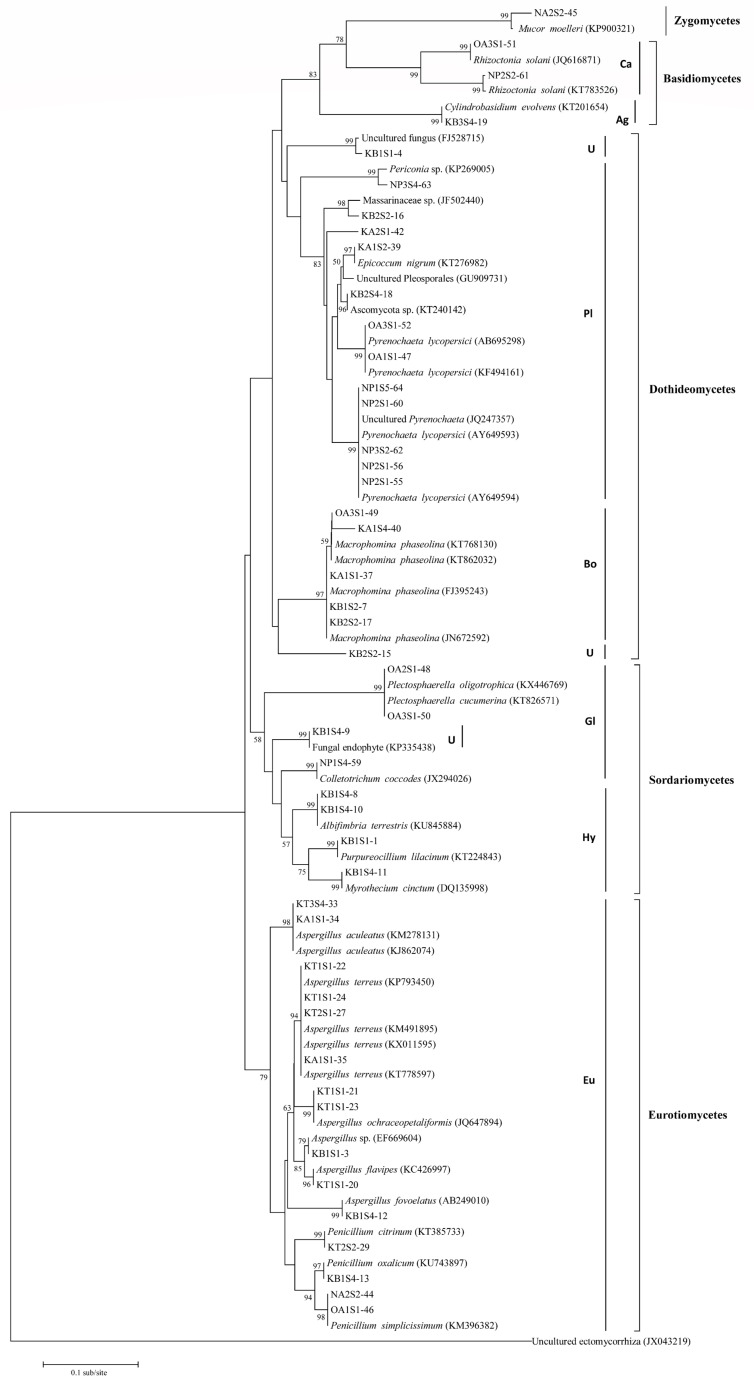
Phylogram of 46 rDNA ITS sequences of root endophytic fungi and their closest BLAST matches based on neighbor-joining analysis. Bootstrap values of >50% are shown at branching points. Ca: Cantharellales; Ag: Agaricales; U: Unknown; Pl: Pleosporales; Bo: Botryosphaeriales; Gl: Glomerellales; Hy: Hypocreales; and Eu: Eurotiales.

**Figure 4 plants-11-01605-f004:**
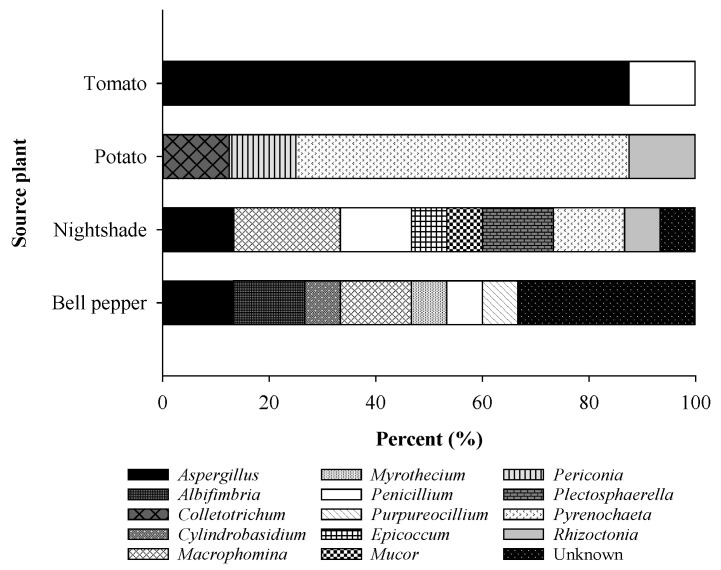
Comparative distribution of taxa with antagonistic activity against *P. infestans* from four solanaceous plant species.

**Figure 5 plants-11-01605-f005:**
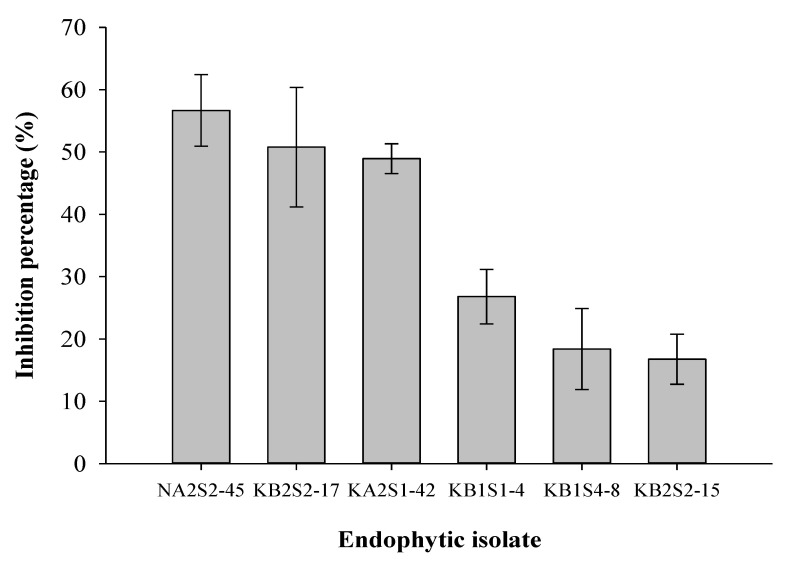
Suppressive effects (%) of the VOCs secreted by the endophytic fungi against mycelial growth of *P. infestans*. Error bars represent the standard error of means (n = 8).

**Figure 6 plants-11-01605-f006:**
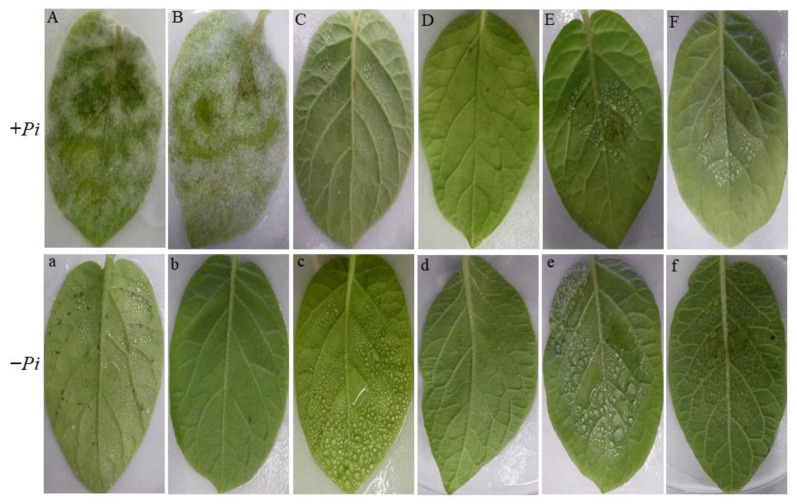
Detached potato leaflets treated or not with crude extracts of the endophytes and inoculated (+*Pi*, uppercase latters) or not (−*Pi*, lowercase letters) with sporangial suspension of *P. infestans*. (**A**,**a**): water control, (**B**,**b**): solvent control, (**C**,**c**): Infinito^®^, (**D**,**d**): KB1S1-4, (**E**,**e**): KB2S1-15, and (**F**,**f**): KA2S1-42.

**Table 1 plants-11-01605-t001:** Molecular identity of endophytic fungi from four solanaceous plants.

Isolates	Accession Numbers	Host Plant	Best BLAST Match (Accession Number)	Identity (%)
KB1S2-7	MG214581	*Capsicum annum*	*Macrophomina phaseolina* (KT862032)	99
NA2S2-45	MG214587	*Solanum nigrum*	*Mucor moelleri* (KP900321)	99
KA1S1-37	MG214583	*S. nigrum*	*Macrophomina phaseolina* (FJ395243)	100
OA3S1-49	MG214585	*S. nigrum*	*Macrophomina phaseolina* (KT768130)	100
KA1S4-40	MG214584	*S. nigrum*	*Macrophomina phaseolina* (KT768130)	100
OA3S1-51	MG214604	*S. nigrum*	*Rhizoctonia solani* (JQ616871)	100
NP2S2-61	MG214605	*S. tuberosum*	*Rhizoctonia solani* (KT783526)	99
KA1S1-34	MG214568	*S. nigrum*	*Aspergillus aculeatus* (KJ862074)	100
KB1S1-3	MG214573	*C. annum*	*Aspergillus* sp. (EF669604)	99
KT1S1-20	MG214569	*Lycopersicon esculentum*	*Aspergillus flavipes* (KC426997)	100
OA1S1-46	MG214592	*S. nigrum*	*Penicillium simplicissimum* (KM396382)	99
KB1S4-8	MG214560	*C. annum*	*Albifimbria terrestris* (KU845884)	99
KB2S2-17	MG214582	*C. annum*	*Macrophomina phaseolina* (JN672592)	100
KB2S4-18	MG214565	*C. annum*	Ascomycota sp. (KT240142)	100
KB1S4-10	MG214561	*C. annum*	*Albifimbria terrestris* (KU845884)	99
KT1S1-21	MG214571	*L. esculentum*	*Aspergillus ochraceopetaliformis* (JQ647894)	100
KT1S1-24	MG214575	*L. esculentum*	*Aspergillus terreus* (KP793450)	99
KT2S1-27	MG214576	*L. esculentum*	*Aspergillus terreus* (KX011595)	100
KB1S4-13	MG214590	*C. annum*	*Penicillium oxalicum* (KU743897)	100
KB3S4-19	MG214579	*C. annum*	*Cylindrobasidium evolvens* (KT201654)	100
KT3S4-33	MG214567	*L. esculentum*	*Aspergillus aculeatus* (KM278131)	100
KT1S1-22	MG214574	*L. esculentum*	*Aspergillus terreus* (KT778597)	100
OA3S1-52	MG214598	*S. nigrum*	*Pyrenochaeta lycopersici* (AB695298)	99
KB2S2-16	MG214586	*C. annum*	*Massarinaceae* sp. (JF502440)	97
KB1S4-9	MG214563	*C. annum*	Fungal endophyte isolate (KP335438)	99
NP1S4-59	MG214578	*S. tuberosum*	*Colletotrichum coccodes* (JX294026)	99
NA2S2-44	MG214591	*S. nigrum*	*Penicillium simplicissimum* (KM396382)	99
KA1S2-39	MG214580	*S. nigrum*	*Epicoccum nigrum* (KT276982)	100
OA2S1-48	MG214595	*S. nigrum*	*Plectosphaerella oligotrophica* (KX446769)	99
KB1S4-11	MG214588	*C. annum*	*Myrothecium cinctum* (DQ135998)	100
OA3S1-50	MG214594	*S. nigrum*	*Plectosphaerella cucumerina* (KT826571)	100
KB1S4-12	MG214570	*C. annum*	*Aspergillus foveolatus* (AB249010)	100
KT1S1-23	MG214572	*L. esculentum*	*Aspergillus ochraceopetaliformis* (JQ647894)	100
KA1S1-35	MG214577	*S. nigrum*	*Aspergillus terreus* (KM491895)	99
NP1S5-64	MG214602	*S. tuberosum*	*Pyrenochaeta lycopersici* (AY649593)	99
NP2S1-55	MG214599	*S. tuberosum*	*Pyrenochaeta lycopersici* (AY649594)	100
NP2S1-60	MG214603	*S. tuberosum*	Uncultured *Pyrenochaeta* (JQ247357)	100
KB1S1-1	MG214596	*C. annum*	*Purpureocillium lilacinum* (KT224843)	100
NP3S4-63	MG214593	*S. tuberosum*	*Periconia* sp. (KP269005)	97
KA2S1-42	MG214566	*S. nigrum*	Uncultured *Pleosporales* (GU909731)	94
NP2S1-56	MG214600	*S. tuberosum*	*Pyrenochaeta lycopersici* (AY649594)	100
NP3S2-62	MG214601	*S. tuberosum*	*Pyrenochaeta lycopersici* (AY649594)	100
OA1S1-47	MG214597	*S. nigrum*	*Pyrenochaeta lycopersici* (KF494161)	100
KB1S1-4	MG214562	*C. annum*	Uncultured fungus (FJ528715)	99
KT2S2-29	MG214589	*L. esculentum*	*Penicillium citrinum* (KT385733)	100
KB2S2-15	MG214564	*C. annum*	Uncultured ectomycorrhiza (JX043219)	91

**Table 2 plants-11-01605-t002:** Mycelial growth inhibition of *P. infestans* and inhibition zones caused by endophytic fungi (arranged in descending order in terms of their mycelial growth inhibition) in dual culture.

Isolate	Highest BLAST Affinities	Mycelial Growth Inhibition (%) *	Inhibition Zone (mm) *
T16	*Trichoderma harzianum*	84.5 ^a^	-
KB1S2-7	*Macrophomina phaseolina*	78.8 ^ab^	-
KA1S1-34	*Aspergillus aculeatus*	76.7 ^ab^	-
NA2S2-45	*Mucor moelleri*	73.8 ^ab^	-
KB2S2-17	*Macrophomina phaseolina*	72.0 ^ab^	-
KA1S1-37	*Macrophomina phaseolina*	70.4 ^ab^	-
OA3S1-49	*Macrophomina phaseolina*	70.2 ^ab^	-
KA1S4-40	*Macrophomina phaseolina*	69.2 ^ab^	-
OA3S1-51	*Rhizoctonia solani*	69.2 ^ab^	-
NP2S2-61	*Rhizoctonia solani*	68.3 ^bc^	-
KB1S1-3	*Aspergillus* sp.	68.2 ^bc^	-
OA1S1-46	*Penicillium simplicissimum*	56.3 ^dc^	-
KB2S4-18	Ascomycota sp.	54.1 ^de^	-
KT1S1-23	*Aspergillus ochraceopetaliformis*	53.5 ^de^	9.3 ^cd^
KA1S1-35	*Aspergillus terreus*	52.0 ^d–f^	7.0 ^c–e^
KB1S4-10	*Albifimbria terrestris*	51.8 ^d–f^	-
KT1S1-21	*Aspergillus ochraceopetaliformis*	49.5 ^d–g^	-
KT1S1-24	*Aspergillus terreus*	47.0 ^d–h^	-
NP1S4-59	*Colletotrichum coccodes*	46.9 ^d–h^	-
KT2S1-27	*Aspergillus terreus*	46.5 ^d–h^	-
NA2S2-44	*Penicillium simplicissimum*	45.0 ^d–i^	-
NP1S5-64	*Pyrenochaeta lycopersici*	43.9 ^d–j^	10.0 ^cd^
KB1S4-13	*Penicillium oxalicum*	43.5 ^d–j^	-
NP2S1-55	*Pyrenochaeta lycopersici*	42.9 ^e–k^	7.6 ^c–e^
KB1S4-8	*Albifimbria terrestris*	41.5 ^e–k^	-
NP2S1-60	Uncultured *Pyrenochaeta*	39.4 ^f–l^	17.0 ^a^
KB1S1-1	*Purpureocillium lilacinum*	39.0 ^f–l^	6.5 ^de^
KA1S2-39	*Epicoccum nigrum*	39.0 ^f–l^	-
NP3S4-63	*Periconia* sp.	38.4 ^g–m^	11.0 ^bc^
KB3S4-19	*Cylindrobasidium evolvens*	37.3 ^g–m^	-
KA2S1-42	Uncultured Pleosporales	37.0 ^g–n^	18.3 ^a^
NP2S1-56	*Pyrenochaeta lycopersici*	36.5 ^g–n^	11.3 ^bc^
KT3S4-33	*Aspergillus aculeatus*	36.4 ^g–o^	-
NP3S2-62	*Pyrenochaeta lycopersici*	34.5 ^h–o^	7.3 ^c–e^
KT1S1-20	*Aspergillus flavipes*	32.2 ^i–p^	-
OA1S1-47	*Pyrenochaeta lycopersici*	31.4 ^j–q^	8.0 ^c–e^
KT1S1-22	*Aspergillus terreus*	31.1 ^j–q^	-
OA2S1-48	*Plectosphaerella oligotrophica*	30.5 ^k–q^	-
KB1S4-11	*Myrothecium cinctum*	28.1 ^l–r^	-
OA3S1-50	*Plectosphaerella cucumerina*	26.1 ^m–r^	-
OA3S1-52	*Pyrenochaeta lycopersici*	24.7 ^n–r^	-
KB1S1-4	Uncultured fungus	24.2 ^o–r^	18.8 ^a^
KB1S4-12	*Aspergillus foveolatus*	22.3 ^p–s^	-
KB2S2-16	*Massarinaceae* sp.	20.1 ^q–s^	-
KB1S4-9	Fungal endophyte isolate	20.0 ^q–s^	-
KT2S2-29	*Penicillium citrinum*	18.9 ^rs^	4.8 ^e^
KB2S2-15	Uncultured ectomycorrhiza	13.3 ^s^	14.8 ^ab^

***** Means with the same letter are not significantly different at α = 0.05.

**Table 3 plants-11-01605-t003:** Effect of crude extracts from the endophytes on sporangial germination and germ tube elongation of *P. infestans*.

CEs Source	Inhibition (%) ^1^	
	Sporangial Germination ^2^	Germ tube Growth ^3^
KB2S2-15	100.00 ^a^	100.00 ^a^
KA2S1-42	100.00 ^a^	100.00 ^a^
KB1S1-4	69.38 ^b^	82.73 ^b^
NA2S2-45	30.04 ^c^	−1.92 ^e^
Acetone (solvent) control	0.00 ^d^	0.00 ^e^

(^1^) Inhibition percentages calculated relative to the solvent control. Means with the same letter within each column are not significantly different at α = 0.05, (*p* < 0.0001), n (^2^) = 1600, n (^3^) = 160.

## Data Availability

The data presented in this study are all contained within the figures and tables.

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
