# Peer review of "Isolation, Identification, and Biocontrol Potential of Root Fungal Endophytes Associated with Solanaceous Plants against Potato Late Blight (*Phytophthora infestans*)"

_plants, 2022, doi:10.3390/plants11121605_

Round 1
Reviewer 1 Report
The manuscript was written in a manner that conforms to standard scientific method. A reasonable number of references are cited as evidence of substantive background. The topic covered in the paper is interesting from a scientific point of view, as it implements the knowledge in the control of potato light blight, using Biological Control Agents (BCAs), that represent a fundamental topic for the development of a sustainable agriculture from a health and environmental point of view. The methods and experimental procedures are described with enough detail to enable future researchers to follow-up aspects of the authors’ work. The results are presented in a clear manner using graphical and table means. The significance of results obtained was interpreted and discussed with due reference to previously published studies. Based on these considerations, I believe that the article can be published without revisions.
Author Response
Dear Reviewer,
We highly appreciate the efforts made in reviewing our manuscript.
Thank you very much.
Kind regards
Abbas El-Hasan
on behalf of the authors
Reviewer 2 Report
The authors present an interesting manuscript with some new results on the effects of endophytes against the plant pathogen P. infestans. The manuscript is well written and researched, the methods and results are described clearly, and the discussion is adequate. In the last 20+ years many papers on the potential of endophytes as biocontrol agents were published, but only very few commercial products were developed and even less are used widely with good results. Still, it makes sense to search for new strains and new metabolites which could be used as alternatives to chemical pesticides. I recommend publication of the manuscript. The author discussed the need for purification and elucidation of the chemical structures of the bioactive molecules. In addition, I suggest that –for a follow-up publication- the three most interesting endophytic strains should be characterized in more detail, including multi-gene phylogenetics and growth patterns at different temperatures. It should be tested if the strains might sporulate under different conditions, such as different media, temperatures, light conditions etc. The production of VOCs and other bioactive metabolites at different temperatures could also be tested, considering that many interesting endophytes were detected from a very hot environment.
Author Response
Dear Reviewer,
Thanks a lot for taking the time and efforts to review our manuscript and the valuable future research ideas.
We agree that the selected promising endophytes could be adapted for very high temperatures which might push them ahead as potent BCAs.
Thanks again.
Kind regards
Abbas El-Hasan
on behalf of the authors
Reviewer 3 Report
Authors of the reviewed manuscript performed a well-organized and comprehensive study intended to reveal endophytic microorganisms possessing the biocontrol activity towards P. infestans. After a screening of a number of root samples, isolation and identification of associated endophytic species, and their testing for antagonistic activity through a number of assays, authors reported three promising isolates showed high inhibitory activity in both in vitro and in vivo tests. The further investigation of these isolates and substances responsible for the target activity may result in some practical and fundamental findings in the field of organic crop protection.
In my opinion, the paper can be published after correction of several minor uncertainties and mistakes in the graphic presentation of results. Below I listed the corresponding comments.
Abstract
Line 21: I suggest it would be good to specify the way of application of the extract: spraying of leaves?
Results
Fig. 1: what does horizontal line means in the case of “potato” bar? There is no difference between two bar parts divided by this line.
Line 112-113: why did you exclude Fusarium species from the further study? A number of publications describe various endophytic Fusarium strains as beneficial and able to improve some plant characteristics or to control plant diseases. In Materials and methods section you wrote they were excluded as pathogens; at the same time, Rhizoctonia solani is a serious potato pathogen, but you did not exclude it from the study.
Line 122: “only nightshade plants were accompanied by potential antagonists from all three regions”. However, Fig. 2 shows that one of the region (Nyandarua) was presented by potential biocontrol strains from potato and tomato only, i.e., nightshade gave such strains only in two regions. Please, check and correct, if necessary.
Fig. 4: please check the correctness of the data. According to lines 167-168, Pyrenochaeta species dominated among those fungi collected from potato. At the same time, Fig. 4 shows the dominance of Rhizoctonia solani. In the case of tomato, Fig. 4 shows three fungal species, while the text (line 167) describes only two. Bell pepper is described as the host of 7 genera, while Fig. 4 shows only 5 color variants for this bar. No Cylindrobasidium, Periconia, or Epicoccum presence is shown, though they present in the list of fungi of Fig. 4.
Table 2: probably it would be good to list the pathogens in the descending order (in relation to the mycelial growth inhibition) since it would provide the better understanding of the most potent ones.
Author Response
Dear Reviewer,
Thank you very much for reviewing our manuscript and the valuable comments provided.
In the attached file, we give a detailed response to your remarks and questions.
Kind regards
Abbas El-Hasan
on behalf of the authors

Reviewer 4 Report
The authors report an interesting phytopathological study on the potato plant. In particular, the authors isolate about 400 endophytic species from the roots of 4 species of solanaceae. The isolates were identified from a genomic point of view with the attribution of the specific fungal species. Finally, the anti P. infenstans action was evaluated in-vitro and in-planta.
The results are excellent and useful for using fewer and fewer chemicals in agriculture. The statistical approach is correct and the discussion sufficiently thorough.
The only unclear aspect concerns the VOCs. The authors in many parts of the manuscript argue for VOCs but there is no gas-chromatographic characterization that illustrates these aspects.
Author Response
Dear Reviewer,
We highly appreciate the efforts made in reviewing our manuscript.
We estimated the production of bioactive VOCs by the endophytes through a standard bioassay. However, VOCs analysis through chromatographic tools would be a nice aspect for our future work.
Thank you very much.
Kind regards
Abbas El-Hasan
on behalf of the authors